# Drought Characteristics and Its Response to the Global Climate Variability in the Yangtze River Basin, China

**Tao Huang [1,2], Ligang Xu [1,2,3,*] and Hongxiang Fan [1,2]** 

[1] Key Laboratory of Watershed Geographic Sciences, Nanjing Institute of Geography and Limnology, Chinese Academy of Sciences, Nanjing 210008, China; huangtao16@mails.ucas.ac.cn (T.H.); fanhongxiang13@mails.ucas.ac.cn (H.F.)
[2] College of Resources and Environment, University of Chinese Academy of Sciences, Beijing 100049, China
[3] Office of Mountain-River-Lake Development Committee of Jiangxi Province, Nanchang 330046, China
[*] Correspondence: lgxu@niglas.ac.cn; Tel.: +86-025-8688-2105

**Abstract:** The frequent occurrence of drought events in humid and semi-humid regions is closely related to the global climate variability (GCV). In this study, the Standard Precipitation Evapotranspiration Index (SPEI) was taken as an index to investigate the drought in the Yangtze River Basin (YRB), a typical humid and semi-humid region in China. Furthermore, nine GCV indices, such as North Atlantic Oscillation (NAO) were taken to characterize the GCV. Correlation analysis and a joint probability distribution model were used to explore the relationship between the drought events and the GCV. The results demonstrated that there were six significant spatiotemporal modes revealed by SPEI3 (i.e., seasonal drought), which were consistent with the distribution of the main sub basins in the YRB, indicating a heterogeneity of drought regime. However, the SPEI12 (i.e., annual drought) can only reveal five modes. Precipitation Indices and El Niño/Southern Oscillation (ENSO) Indices were more closely related to the drought events. A causal relationship existed between ENSO precipitation index (ESPI), NAO, East Central Tropical Pacific Sea Surface Temperature (Nino3.4) and Northern Oscillation Index (NOI) and drought in the YRB, respectively. Drought events were most sensitive to the low NAO and high NOI events. This study shows a great significance for the understanding of spatiotemporal characteristics of meteorological drought and will provide a reference for the further formulation of water resources policy and the prevention of drought disasters.

**Keywords:** SPEI; the GCV; joint probability distribution; the Yangtze River basin; drought

---

## 1. Introduction

Drought is a serious natural disaster all over the world. It is often considered to be one of the well-known natural disasters [1–4]. The effects of drought are often magnified especially in the context of global climate change [5–8]. Generally, drought can be divided into four types: Meteorological drought, Agricultural drought, Hydrological drought and Social-economic drought [9]. Meteorological drought is a prerequisite for the other three droughts. Droughts occur over most parts of the world, both in arid and humid regions [10]. Understanding the spatiotemporal variations of drought is of primary importance for freshwater planning and management [11].

Drought regimes all over the world have changed dramatically in recent years [12]. This variability was often linked to both internally and externally forced fluctuations in the global Sea Surface Temperature (SST) field [13]. El Niño/Southern Oscillation (ENSO) is a coupled ocean-atmosphere

tropical Pacific phenomenon worldwide. A number of studies have confirmed that there were connections between ENSO and regional climate extremes like floods and droughts [14]. A concern exists that the ongoing global warming may increase the severity of droughts and many studies have shown that drought severity increases with temperature rise [10,15–18]. For instance, from 1972 to 2004, the warming increased global dry areas by 20% to 38% [15]. The Intergovernmental Panel on Climate Change (IPCC) Fourth Assessment Report [19] proposed a continuous increase in global mean surface air temperature during the 21st century owing to the increase of anthropogenic greenhouse gas concentration [20], indicating an increase in drought frequency, severity and spatial extension in future.

Drought can be characterized by many indices such as the Palmer Drought Severity Index (PDSI), Standardized Precipitation Index (SPI) and Standardized Precipitation Evapotranspiration Index (SPEI) [21–23]. PDSI and SPI have a wide range of applications, but each of them has its limitations. PDSI takes account of changes in surface water balance so that it is suitable for the measurement of long-term drought conditions, but the acquisition is difficult because the calculation process is complex, and the time scale is limited [24]. SPI takes precipitation as a factor. Although its calculation method is simple and time scale is flexible, it does not consider temperature, evapotranspiration and other influencing factors [25–28]. The SPEI is a site-specific drought indicator of deviations from average water balance (precipitation minus potential evapotranspiration). One of the weaknesses is that it requires more data than the SPI. Similar to the SPI, the SPEI also has trouble dealing with arid climates where precipitation is near zero [28]. However, the SPEI includes the effect of temperature on drought severity by means of atmospheric evaporation demand. In summary, SPEI is widely used in research on drought.

Flexible Copula function is one of the most popular methods used in multivariate frequency studies of hydrological droughts [29–32]. Previous studies usually aimed at investigating the joint probabilities of different characterizations for a single hydrological drought event [33]. For instance, Zhang and Singh [32] investigated bivariate rainfall frequency distributions by using Archimedean Copulas. The advantage of the Copula is that it can construct a joint probability distribution between any two variables without considering the marginal distribution of each variable. Normally, the Copula method was applied in the research of different drought characteristics like duration and severity [31–34]. Considering the independence of the marginal distribution, it can be used to explore the probability of the co-occurrence of two variables that may have potential connections. In this way, the statistical relationship between the drought and the GCV can be established.

Due to the frequent droughts in recent years, the Yangtze River Basin (YRB) had attracted more and more attention from researchers [35]. The trend of runoff and precipitation has been analyzed and the impacts of changes has been discussed using ground-based meteorological data [36]. Many studies also showed that in the lower and middle Yangtze catchments, El Niño showed a close relation with flood events and La Niña correlates with drought events [37]. This indicated that drought /flood in the YRB was specific and easy to change with the global climate variability (GCV). This study treats the YRB as a typical area that has a complex regime of drought and waterlogging, and an area which is influenced by GCV sensitively. Meanwhile, nine typical GCV indices were selected to characterize the GCV. The objectives of this study are as follows. (1) Examine the spatiotemporal characteristics of droughts patterns (or regimes) in the YRB during 1960–2013, and examine the trend of SPEI in different time scales. (2) Discuss the relationship between SPEI and nine typical GCV indices. (3) Quantitatively characterize the statistical relationship between the drought and the GCV. Studies on drought in the YRB were mainly focused on the hydrological drought, and there was not much research on the meteorological drought, which was most susceptible to the GCV. From the perspective of quantitative statistics, the research on the correlation of drought in the YRB mostly focused on the qualitative description. It is necessary to predict the probability of drought in a quantitative way. The results may improve the understanding of drought regimes in the YRB and the relationship with the GCV, which is helpful and meaningful for the water resource management and the social production of the YRB.

## 2. Study Area and Data

The Yangtze River Basin (24~35° N, 90~122° E) (Figure 1) flows approximately 6380 km from a glacier on the Qinghai-Tibetan Plateau eastward across south-west, central, and eastern China, finally draining in the East China Sea [38]. The Yangtze River has a drainage area of $1.8 \times 10^6$ km$^2$, occupying 20% of China's total landmass. The river basin accounts for 36% of China's water resources and approximately 45% of China's total gross domestic product (GDP) [39]. The basin has a gently sloping topography that drops from above 5000 m to sea level. Most sections of the basin are situated within a subtropical warm-wet zone and affected heavily by monsoon climate; however, the Yangtze River source region is in the dry, frigid, high-altitude zone [40]. The annual precipitation ranges from 500 mm in the west to 2500 mm in the east, and more than 60% of it is concentrated in summer (June, July and August). Alternating seasonal air-mass movements and accompanying winds result in humid summers and dry winters [41].

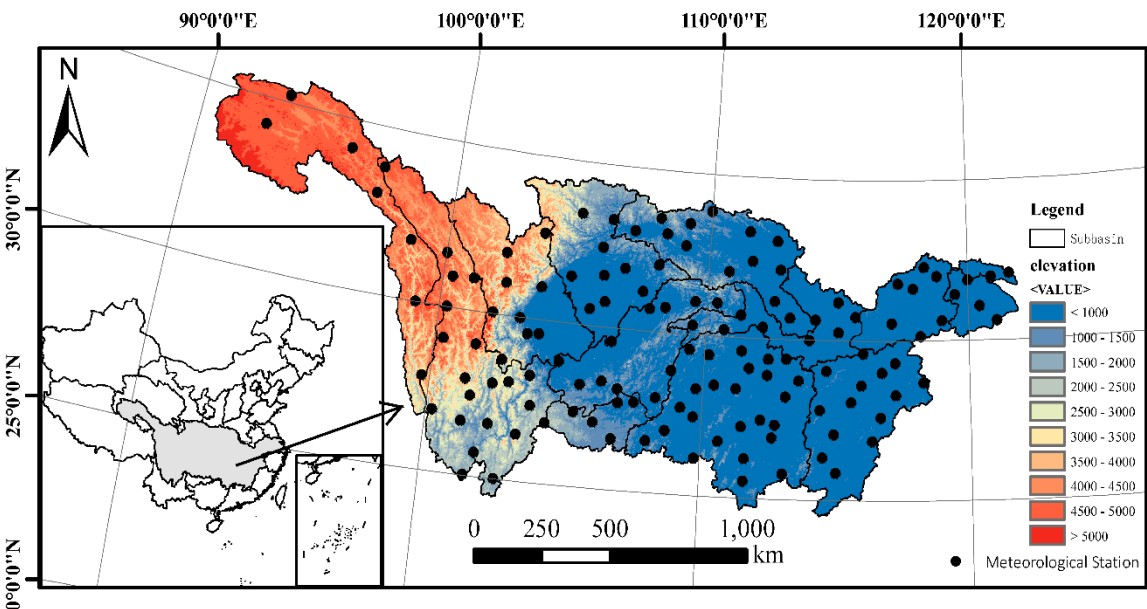

**Figure 1.** The Yangtze River basin.

Data from 137 National Meteorological Observatory (NMO) stations including daily observations of maximum, minimum and mean air temperature, precipitation, wind speed, relative humidity, sunshine hours, for the period of 1960–2013 were used in this study. They have been provided by the National Climatic Centre (NCC) of CMA (the China Meteorological Administration). The daily records of meteorological variables provided by CMA had gone through a standard quality control process before delivery, and with no missing data on the variables used in this study [42]. For ENSO data, all the GCV indices are listed in Table 1. The time series of PDO, SOI, et al., is provided by the Earth System Research Laboratory [43].

**Table 1.** Description of the global climate variability (GCV) indices used in the study. NOAA is the abbreviation of National Oceanic and Atmospheric Administration.

| Abbreviation | Index Name | Period | Data Source |
|:---:|:---:|:---:|:---:|
| AAO | Antarctic Oscillation | 1979–2016 | NOAA |
| ESPI | ENSO precipitation index | 1979–2016 | NOAA |
| NAO | North Atlantic Oscillation | 1948–2001 | NOAA |
| Nino3.4 | East Central Tropical Pacific SST | 1948–2017 | NOAA |
| NOI | Northern Oscillation Index | 1948–2007 | NOAA |
| PDO | Pacific Decadal Oscillation | 1948–2017 | NOAA |
| SOI | Southern Oscillation Index | 1948–2017 | NOAA |
| TNA | Tropical Northern Atlantic | 1948–2017 | NOAA |
| TSA | Tropical Southern Atlantic | 1948–2017 | NOAA |

## 3. Methods

### 3.1. SPEI

The Standardized Precipitation Evapotranspiration Index (SPEI) is a relatively new drought index, which includes both precipitation and potential evapotranspiration (PET) influence. A previous study admitted that there was a difficulty with arid climates where precipitation is near zero, considering that the SPEI is sensitive to precipitation [28]. Due to the humid climate in the YRB, the SPEI is suitable in this study. The empirical Thornthwaite equation [44] and the Penman–Monteith equation are the most popular methods [45,46] used to calculate PET. The empirical Thornthwaite equation is considered to be most convenient because the air temperature is the only variable in the formula. For the assessment of meteorology drought, this method provides a fast way to obtain a PET. When there are more observation data, such as wind speed, cloud amount and relative humidity etc., the Penman–Monteith equation is a better choice due to its consideration of energy balance and atmospheric water demand [47]. PET was estimated following the Penman–Monteith method recommended by FAO56 (The Food and Agriculture Organization Irrigation and Drainage paper 56 document) in this study. For more details, the calculator handbook is recommended [48]. Computation of SPEI is based on monthly (or weekly) difference between precipitation and PET, representing a climatic water balance, at the different time scale of interest. A package of R software [49] named SPEI can be used to compute this index. For specific function and the relevant code, the website of this package is available [50].

### 3.2. EOF/REOF Method

Rotation Empirical Orthogonal Function (REOF) is a statistical and exploratory tool that can identify and extract spatiotemporal patterns of geophysical variables [51]. The EOF analysis involves three components: Eigenvectors (EOFs, SPs), principal components (time coefficients) and eigenvalues. The varimax REOF method is proposed to optimize and simplify local structures. The REOF patterns based on Varimax can be obtained by altering the EOF structures. The r dominant EOFs, $a_{m \times r}$ matrix $a_{m \times r} = (a_1, a_2, \ldots, a_r)$, are used to construct the REOFs $W$ by means of an $r \times r$ rotated matrix $R$. The relationship between EOFs and REOFs can be expressed as follows:

$$W = a_{m \times r} \times R_{r \times r}, \tag{1}$$

where the rotated matrix $R$ meets the condition $R^T R = I$. The matrix $I$ is the unit matrix. Moreover, the rotated matrix $R$ can be obtained according to a rotation criterion [52].

### 3.3. Wavelet and Lag Analysis

Based on the two Continuous Wavelet Transforms (CWTs), we can construct the Cross Wavelet Transform (XWT) to expose their common power and relative phase in time-frequency space. For two time series $X$ and $Y$:

$$W_n^{XY} = W_n^X(s)W_n^{Y*}(s), \tag{2}$$

where $W_n^X(s)$ and $W_n^Y(s)$ are CWTs of $X$ and $Y$, respectively. "*" represents complex conjugation. $S$ represents the scale of expansion. $W_n^{XY}$ is the power spectrum of XWT.

Wavelet Transform Coherence (WTC) is another tool to find how coherent the cross-wavelet power is in time-frequency space:

$$R_n^2(s) = \frac{\left| S\left(s^{-1}W_n^{XY}(s)\right)\right|^2}{S\left(s^{-1}|W_n^X(s)|^2\right) \times S(s^{-1}|W_n^Y(s)|^2)}. \tag{3}$$

Even if the corresponding region of the low energy value of the XWT power spectrum is small, the correlation in the wave coherent spectrum is also likely to be significant. This definition is close to the traditional correlation coefficient formula and easily understood.

Time series of the SPEI and the GCV indices were examined to check the assumption of normal distribution using the Kolmogorov–Smirnov test. None of the tested time series were normally distributed. Accordingly, Spearman's and Kendall's rank correlation coefficients were calculated to assess the relationship between the drought and the GCV.

### 3.4. Copulas

Considering a situation with two random variables, Sklar's Theorem states that if $F_{X,Y}(x,y)$ is a two-dimensional distribution function with marginal distributions $F_X(x)$ and $F_Y(y)$, then there exists a Copula $C$ such that

$$F_{X,Y}(x,y) = C(F_X(x), F_Y(y)). \tag{4}$$

For any univariate distributions $F_X(x)$ and $F_Y(y)$ and any Copula $C$, the function $F_{X,Y}(x,y)$ defined above is a two-dimensional distribution function with marginal distributions $F_X(x)$ and $F_Y(y)$. Furthermore, if $F_X(x)$ and $F_Y(y)$ are continuous, then $C$ is unique. Under the assumption that the marginal distributions are continuous with probability density functions $f_X(x)$ and $f_Y(y)$, the joint probability density function then becomes

$$f_{X,Y}(x,y) = C(F_X(x), F_Y(y)) \times f_X(x) \times f_Y(y), \tag{5}$$

where $f_{X,Y}(x,y)$ is the density function of $C$, defined as Joe [53], Frees and Valdez [54], Demarta and McNeil [55], Cherubibi [56] provided a number of one-parameter families of Copulas. In this study, four Copulas like Ali-Mikhail-Haq, Clayton, Frank and Gumbel-Hougaard were tested to see whether they could fit the distribution well. The corrected Akaike Information Criterion was applied to test the goodness of fit:

$$AICc = -2\ln(\hat{L}) + 2k + \frac{2k(k+1)}{n-k-1}, \tag{6}$$

where $k$ is the number of parameters in the corresponding distribution, $n$ is the sample size, and $\hat{L}$ is the maximum value of the likelihood function for the distribution. A smaller $AICc$ value implies a better goodness-of-fit.

According to the relationship between marginal probability distribution and joint probability distribution, the probability when the GCV is smaller than the threshold value is $C(F_X(x), F_Y(y))$ itself. When the GCV is larger than the threshold, the probability is

$$F_{X,Y}(x,y) = F_X(x) - C(F_X(x), F_Y(y)). \tag{7}$$

## 4. Results and Discussion

### 4.1. Spatial Characteristics of the Drought-Wet Regime in the YRB

The first six modes of REOFs of SPEI3 and SPEI12, which represent a 3-month-scale of SPEI (a 3-month moving average) and a 12-month-scale SPEI (a 12-month moving average) respectively, were used to describe the spatial characteristics of the drought in the YRB. For both SPEI3 and SPEI12, the cumulative variance of their first six rotated principal components is about 60%, which shows that they can represent the characteristics of drought well in general. The six principal components after the rotation of the SPEI3 can clarify the obvious spatial differences. In Figure 2, the first mode shows that the high-loading area is the south-west part of the YRB, belonging to the upper reaches of the Yangtze River, containing the Jinsha River, Yalong River. The second mode shows that the north-east part of the YRB has high loadings. This area is the lower reaches of the YRB. It contains the Yangtze River Delta and some parts of the province of Anhui and Jiangxi. Abundant rainfall, which is strongly influenced by the summer monsoon makes the area humid. Notably, the drought we mentioned in the humid and semi-humid regions generally refers to a drought event or a seasonal water shortage, which is different from the drought in the arid region like north-west China. In addition, the high-loading region in mode 2 is where drought-flood abrupt alternation happens frequently, according to previous research [57]. REOF1 and REOF2 reveal the heterogeneity of the dry and wet extent between the east and west YRB. REOF3 also represents this difference but its high-loading region is in the south-east of the YRB. The biggest freshwater lake in China is located in this region: The Poyang Lake. The relationship between this lake and the Yangtze River is very complicated so that the drought in this region may be easily influenced by the variation of the Lake-Basin-River system [58]. REOF3 and REOF6 are also regarded as the influence of altitude on dry and wet because the north-west part of the YRB is the Tibet Plateau, which is the origin of the Yangtze River.

It can be seen from REOF4 and REOF5 that there was a difference between the south and north part of the YRB. In the 1950s, since the El Niño period started, 16 El Niño events have occurred in the years 1950–1998 [59]. The variation of the global sea surface temperature in the Pacific Ocean was strongly related to the drought and flood disasters in the spring and summer of the YRB in China. In the frequent years of El Niño, summer rainfall in China has been characterized as Wet in the south and Drought in the north, which means more precipitation in the south and less rainfall in the north [60]. When El Niño happened, it was prone to rainfall less in the north than the south. In summary, the results of SPEI3 data decomposed by REOF fully demonstrated the characteristics of the complex drought distribution in the YRB, considering huge differences caused by longitude and latitude. Moreover, El Niño plays an important role in the drought occurrence in the humid and semi-humid region influenced by the summer monsoon frequently.

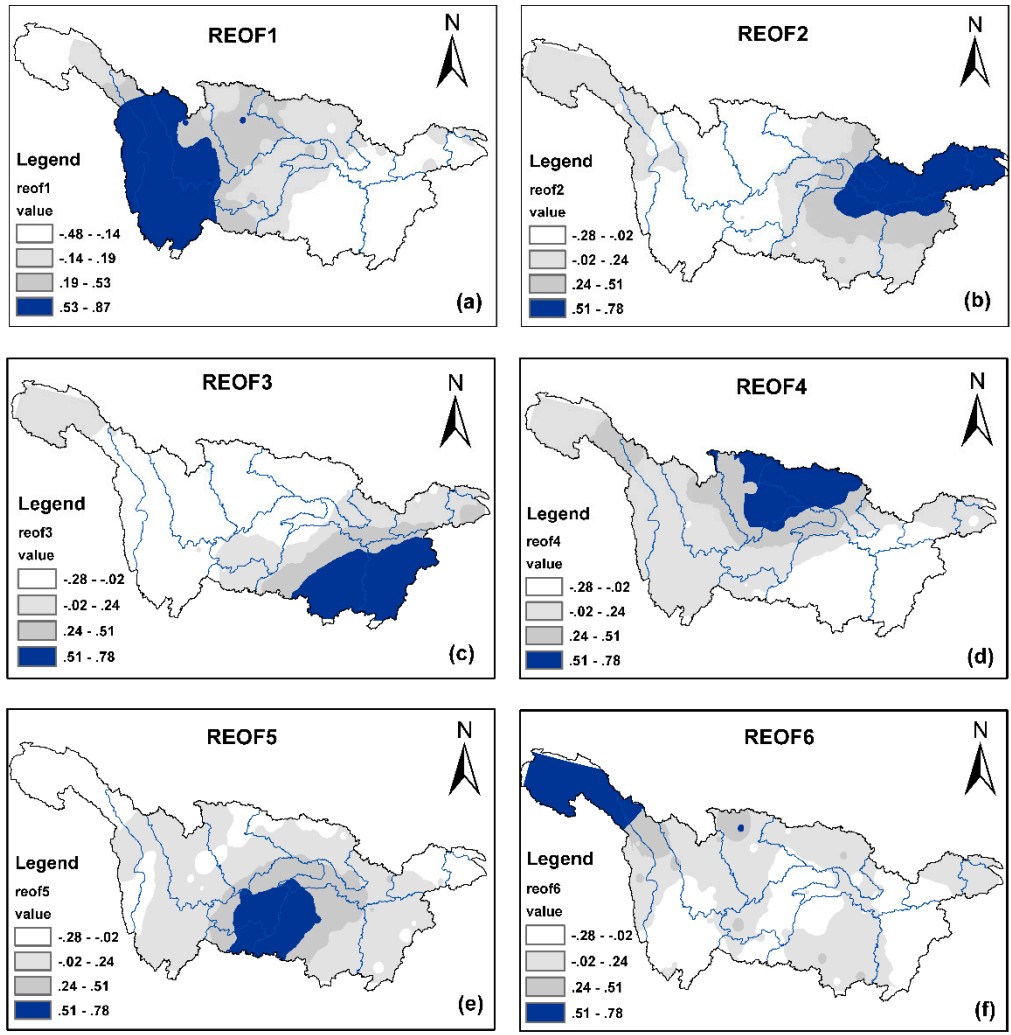

**Figure 2.** The Rotation Empirical Orthogonal Function (REOF) results of the Standard Precipitation Evapotranspiration Index (SPEI) (at 3-month scale). Mode 1 (**a**), Mode 2 (**b**), Mode 3 (**c**), Mode 4 (**d**), Mode 5 (**e**) and Mode 6 (**f**) are the first six modes. The blue area represents the high loading value. The value of the load is divided into four classes. In order to highlight the difference, the rest of the three lower values are expressed in grayscale.

However, the results revealed by SPEI12 were different. As shown in Figure 3, despite the order, the first five modes obtained by SPEI3 can also be described by SPEI12. For the last mode, the high-loading region was not obvious. Compared to SPEI3, SPEI12 cannot accurately reflect the spatial characteristics of the north-west region of the YRB. There are two reliable explanations for this result. For both explanations, the time scale of the SPEI must be taken into consideration. First, the 3-month time scale SPEI can be considered to represent the seasonal variation. In previous research, many studies have shown that ENSO has a seasonal impact on precipitation [61]. The 3-month timescale can reflect the characteristics of the monsoon area in detail because the seasonal precipitation varies greatly. Second, the Tibet Plateau is a cold region with high altitude. SPEI itself is an index that takes both precipitation and evapotranspiration into account. Due to the scarce meteorological stations in the north-west of the basin, and the interannual characteristics SPEI12 can only reflect the interannual variation of evapotranspiration. Therefore, we can conclude that in the humid and semi-humid climate region, SPEI3 can better reflect seasonal scale changes. For interannual variability, SPEI12 is also available. Furthermore, if the variability at the monthly and seasonal scales is to be eliminated in order to study interannual changes, the SPEI12 should be the most appropriate.

The following research in this study is based on interannual variability, so the SPEI12 data will be used for the next step.

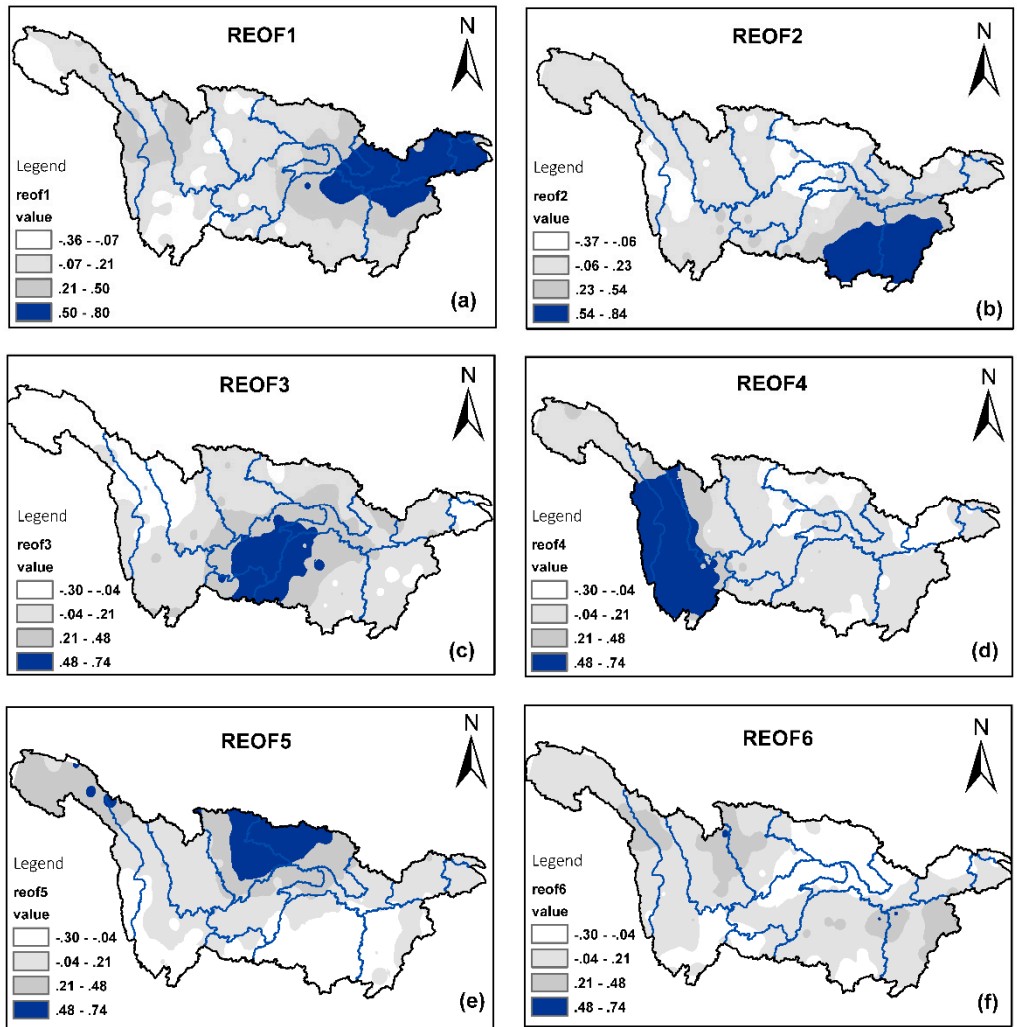

**Figure 3.** The REOF results of SPEI (at 12-month scale). Mode 1 (**a**), Mode 2 (**b**), Mode 3 (**c**), Mode 4 (**d**), Mode 5 (**e**) and Mode 6 (**f**) are the first six modes. The blue area represents the high loading value. The value of the load is divided into four classes. In order to highlight the difference, the rest of the three lower values are expressed in grayscale.

### 4.2. The Correlation Relationship between Drought and the GCV

In order to further study the relationship between drought and the GCV, the correlation analysis of SPEI and nine GCV indices were conducted in this study. The correlation analysis of the time and frequency domain can be accomplished with Cross Wavelet Transform (XWT) and Wavelet Transform Coherence (WTC). The annual average value of SPEI12 was used as the annual average SPEI value. The results were shown in Figures 4 and 5. SPEI and AAO had no common period on the power spectrum of XWT, but they showed a negative correlation during 1983 to 1987, with a common period (2–4 years). There was a significant common period (3–5 years) between SPEI and ESPI both in XWT and WTC during 1991 to 2001. Meanwhile, ESPI was 5/8 phase ahead of SPEI, which means that ESPI was about 5–7 months ahead of SPEI during 1991 to 2001. There was a strong period (8–10 years) between SPEI and NAO from 1976 to 1992, and a strong positive correlation between them as well. In the high-frequency area, there was a period (2–3 years) from 1968 to 1971, and a period (3–4 years) from 1976 to 1985. However, they did not appear in WTC, which means the correlation was not that

significant. The common period of SPEI and Nino3.4 appeared in the high-frequency area, and the strongest periodicity occurred during 1968 to 1974, with a period of 1–4 years. During this stage, it can be observed from the power spectrum of WTC that there was almost a negative correlation between them. The most condensed period (2–4 years), appeared from 1968 to 1974 and was also in the high-frequency region when it comes to NOI. In this period, NOI leads the SPEI 1/4 phase, that is, 3 months ahead of SPEI. Besides, it is worth noticing that although it is not shown on the XWT, we can see a common period with anti-phase from WTC after 2000. PDO and SPEI did not show a common periodicity before 1995, but there was a significant periodicity in the WTC power spectrum with a positive correlation. After 1995, the high-frequency region no longer had a strong common period. There was a common negative correlation periodicity (9–10 years) in the low-frequency area during 1995 to 2000. Among the three common periods obtained by analyzing the relationship between SPEI and SOI, the strongest period showed the same-phase variation tendency. In the other two common periods, SOI was leading SPEI by about 1/8 phases, which was about one and a half months. No significant correlation was observed from the WTC power spectrum of TNA, but TNA and SPEI had a common period from 1976 to 1995. For TSA, after 1995, there was a significant positive correlation area in the high-frequency area. In the period of 4–5 years, the correlation relationship between TSA and SPEI was positive. In the period of 1–2 years that occurred after 2000, TSA was 1/8 phase ahead of SPEI.

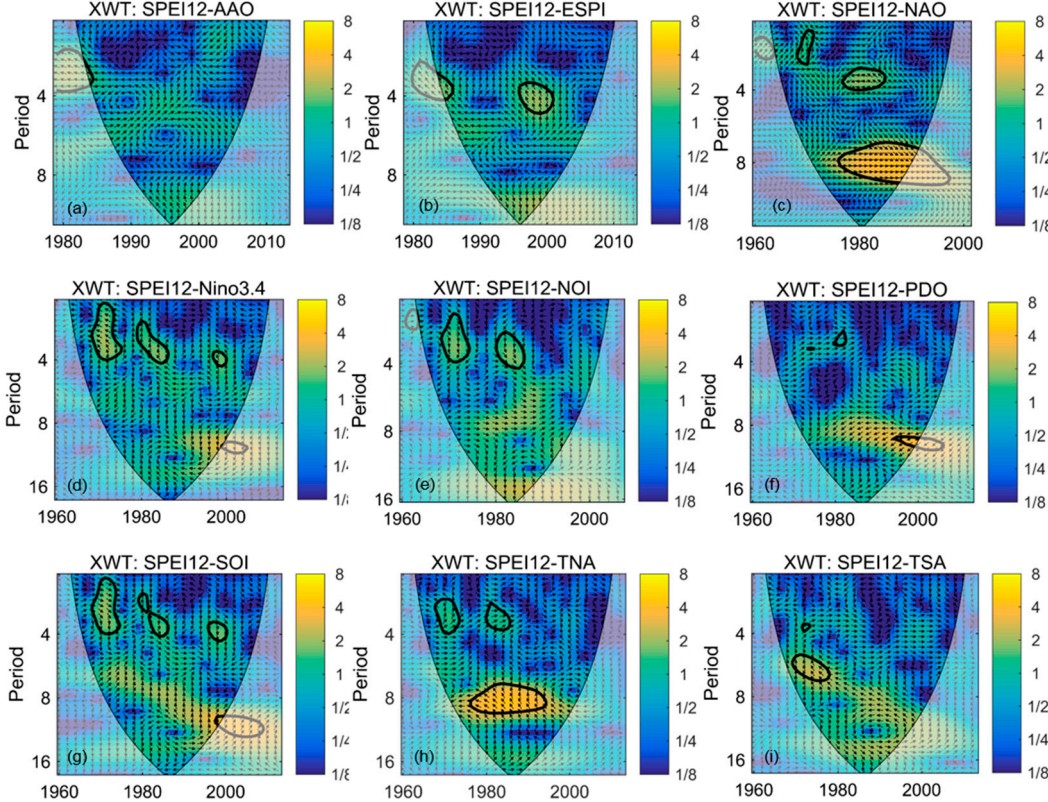

**Figure 4.** The results of Cross Wavelet Analysis (XWT) of SPEI (at 12-month scale) and the GCV.

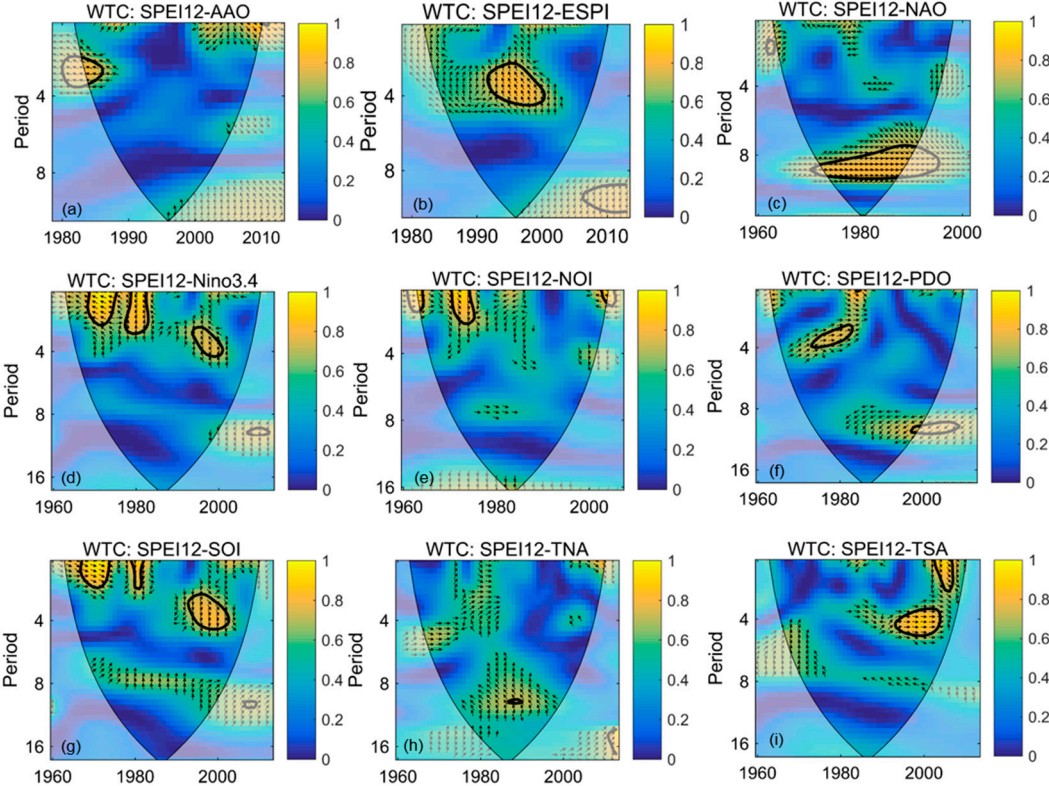

**Figure 5.** The results of Wavelet Transform Coherence (WTC) of SPEI (at 12-month scale) and the GCV.

The GCV indices used in this study were divided into five different types by NOAA, which were: Atmosphere Indices (AAO, SOI), Precipitation Indices (ESPI), Teleconnection Indices (NAO, NOI, and PDO), ENSO Indices (Nino3.4) and Atlantic SST Indices (TNA, TSA). According to the results derived from XWT and WTC, in Atmospheric Indices SOI is more correlated with SPEI than AAO. Among these indices, there was a positive correlation between NAO and SPEI, which means the interannual correlation between NOI and SPEI was better. The interannual correlation between PDO and SPEI was not strong, but the place where there was a common period was also anti-phase. The study of PDO may have to pay more attention to the monthly or seasonal scales PDO, because Xiao, Zhang and Singh [60] showed that a PDO event at the same year tended to increase the spring precipitation in the south-western part of the YRB, which was consistent with the results of Chan and Zhou [59] that the early summer (May to June) monsoon rainfall over South China was related to PDO, with drier (wet) monsoon years during the periods of high (low) PDO indices. The Precipitation Indices and ENSO Indices were most closely related to SPEI. This showed that there was a close relationship between the ENSO events and the precipitation in this region. Compared with TSA, TNA was more associated with the SPEI in the YRB.

It was clear that there were different extents of correlation between SPEI and GCV indices over the past 54 years using correlation analysis. In order to reveal the possible causal relationship between the GCV and the drought, the Pearson linear correlation coefficient is adapted to measure the correlation between them. On the basis of the original Pearson linear correlation method, SPEI indices were lagged for one year and then the correlation coefficient was recalculated. If the correlation was significant, it indicated that this lag was meaningful, and there might be a simple causal relationship between them. The results are shown in Figure 6.

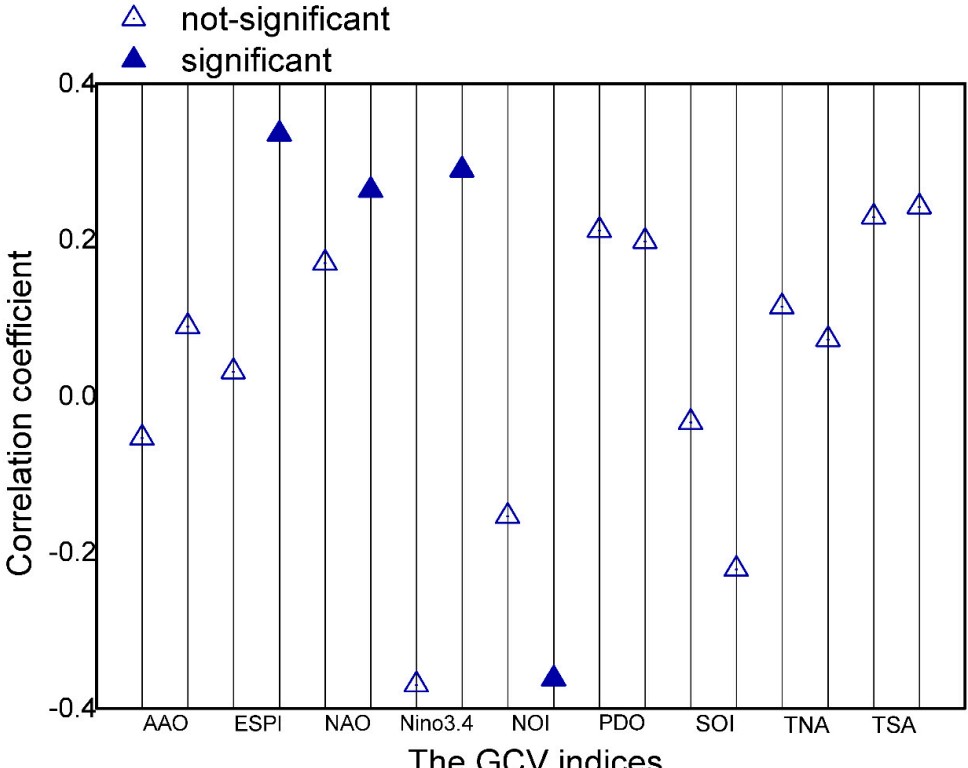

**Figure 6.** The results of lag analysis. The blue hollow triangle represents a not-significant correlation coefficient. The blue solid triangle represents a significant correlation coefficient after a one-year lag. ($p < 0.05$).

First, the correlation relationship between the GCV and the SPEI was not very significant. This suggested that, in general, SPEI did not have a very good immediate response to the GCV. Next, from the result of the lag analysis, when SPEI was postponed for one year, four significant correlation coefficients can be derived from ESPI, NAO, Nino3.4, and NOI. The results implied that there was a certain causal relationship between ESPI, NAO, Nino3.4 and NOI and the drought respectively in the YRB.

ESPI was the ENSO precipitation index derived from time series that used rainfall data in the Tropical Pacific to describe ENSO events. The precipitation pattern of the YRB was changed by ENSO events, resulting in the dry and wet situation change as well. As an index that was also affected by ENSO events, the relationship with SPEI can help us predict the trend of the variation in SPEI. NAO was an important factor affecting the westerly wind in the middle latitudes. Early studies suggest that NAO had obvious local characteristics, and its influence was mainly concentrated in the North Atlantic and its surrounding regions. However, Hurrell [62] pointed out that NAO also had an important impact on climate change in the northern hemisphere. The mid-latitude westerly anomaly caused by NAO can adjust the atmospheric circulation in the plateau area, and then cause the plateau ridge system to strengthen (weaken) at the same time, and eventually cause the formation of the summer plateau precipitation bipolar oscillation mode. On this basis, the dynamic mechanism of the summer plateau precipitation bipolar oscillation mode was further explored by Liu and Yin [63]. The research showed that along with the energy transmission from the North Atlantic to the Asian region, there was an anomaly center related to summer NAO, which was related to the summer NAO in Western Europe, the Mediterranean plateau, the western plateau and East Asia. This series of anomaly centers was the link between the NAO and the Plateau Summer precipitation. NAO adjusts the atmospheric circulation in the plateau and its surrounding areas through the structure of the wave column, which has an important influence on the formation of the bipolar oscillation mode of summer

precipitation in the plateau. It has been previously mentioned that the variation of heat and land differences on China's land and sea caused by the Qinghai-Tibet Plateau will have an impact on the drought in the YRB. Due to the above effects of NAO on the Asian plateau precipitation, it is bound to change the precipitation pattern of the YRB, thus affecting the occurrence of drought events.

There was a lot of research on the impact of ENSO on East Asia and China's summer monsoon. Baohua and Ronghui [64] believed that convective activities in the Pacific and east of the equator were strengthened, while those in the tropical western Pacific were weakened. Southward deflection and westing of subtropical anticyclones created the convective activity of the Equatorial Pacific Ocean on the abnormal formation of a dipole structure in the mature period of the ENSO event. It was obvious that the anomalous heat source over the equatorial Pacific would also form a dipole structure. This thermal anomaly structure was conducive to the forcing of the anomalous circulation in the tropical western Pacific and the South China Sea, and the south-west monsoon circulation in South China was strengthened. Based on this view, the most recent two years of flood season in the YRB in 1998 and 2010 were also the high-value years of SST in the Nino3.4 area. Therefore, it was speculated that ENSO might have an impact on the floods in the south of the Yangtze River. NOI is the only index with a negative correlation. Most of the studies on NOI precipitation in China were concentrated on the microscopic scale, such as seasonal precipitation. There were not many studies on the interannual scale. NOI was an oscillation relative to SOI. This is because it was in the northern hemisphere and also closely related to the climate pattern of China and the YRB in the northern hemisphere.

Restricted by current limited knowledge on underlying mechanisms, statistical correlation methods are often used rather than a physical based model to characterize the connections. In the following study, a Copula method was used to establish a statistical model of the drought events in the YRB and the GCV.

### 4.3. The Quantitative Relationship between Drought Events in the YRB and Global Change

In the study of the joint probability distribution of bivariate random events, the Archimedean Copula function was generally selected [65]. The Ali-Mikhail-Haq, Clayton, Frank and Gumbel-Hougaard Copula functions were used to investigate the four GCV indices which were most closely related to the drought events in the YRB. Normally, one of the crucial things was that the most suitable Copula function between the GCV and the SPEI in the YRB has to be found respectively. For this purpose, the correlation coefficient tau was obtained using the rank correlation analysis. Then, the parameter theta of the Copula function was derived according to tau [66]. The results of the goodness of fit are shown in Table 2.

**Table 2.** The results of the goodness-of-fit values of four Copula functions for ESPI, NAO, Nino3.4 and NOI. **means that no such value exists because the rank correlation is negative.

| Copula | Parameters | SPEI-ESPI | SPEI-NAO | SPEI-Nino3.4 | SPEI-NOI |
|---|---|---|---|---|---|
| | tau | 0.01 | 0.16 | −0.01 | -0.09 |
| GH | theta | 1.01 | 1.20 | ** | ** |
| | *p*-value | 0.67 | 0.27 | ** | ** |
| | tau | 0.01 | 0.16 | −0.01 | −0.09 |
| Clayton | theta | 0.02 | 0.39 | ** | ** |
| | *p*-value | 0.96 | 0.16 | ** | ** |
| | tau | 0.01 | 0.16 | −0.01 | −0.09 |
| AMH | theta | 0.05 | 0.61 | −0.03 | −0.48 |
| | *p*-value | 0.94 | 0.17 | 0.66 | 0.64 |
| | tau | 0.01 | 0.16 | −0.01 | −0.09 |
| FRANK | theta | 0.11 | 1.51 | −0.07 | −0.76 |
| | *p*-value | 0.97 | 0.35 | 0.69 | 0.84 |

It can be found from the results in Table 2 that GH Copula had the best fitting effect on SPEI-ESPI. In the fitting of SPEI-NAO, the performance of Clayton Copula was better than the other three Copulas.

For Nino3.4 and NOI, the GH Copula and Clayton Copula Functions were not applicable because the *tau* value obtained between them and SPEI was negative. For AMH and Frank Copula, SPEI-Nino3.4 and SPEI-NOI both obtained a smaller *p*-value in the AMH Copula. Therefore, AMH Copula was selected to fit SPEI with NINO3.4 and NOI. The margin distributions of these five indices were detected and the results indicated that SPEI, ESPI, NAO, Nino3.4, and NOI all followed the logistic distribution. The location and scale of each of them were shown in Table 3. The purpose of establishing the joint probability distribution was to explore the response of flood-drought events to extreme GCV. The 1/4 and 3/4 quantiles were used to define the extreme GCV. When the SPEI was less than 0, a drought event can be considered. When the value of SPEI was less than −0.5, a relatively obvious drought can be considered. Determining the joint probability of ESPI, NAO, Nino3.4 and NOI when they were smaller than the number of 1/4 quantiles and bigger than 3/4 quantiles and at the same time when SPEI was less than 0 or SPEI was less than −0.5, respectively. The joint probability was used as a quantitative description of the relationship between drought events in the YRB and the GCV. The results when the GCV was smaller than its 1/4 quantiles are in Figures 7 and 8. When the GCV was larger than its 3/4 quantiles, the probability can be obtained according to the statistical method mentioned in Section 3.4. Details can be found in Table 4.

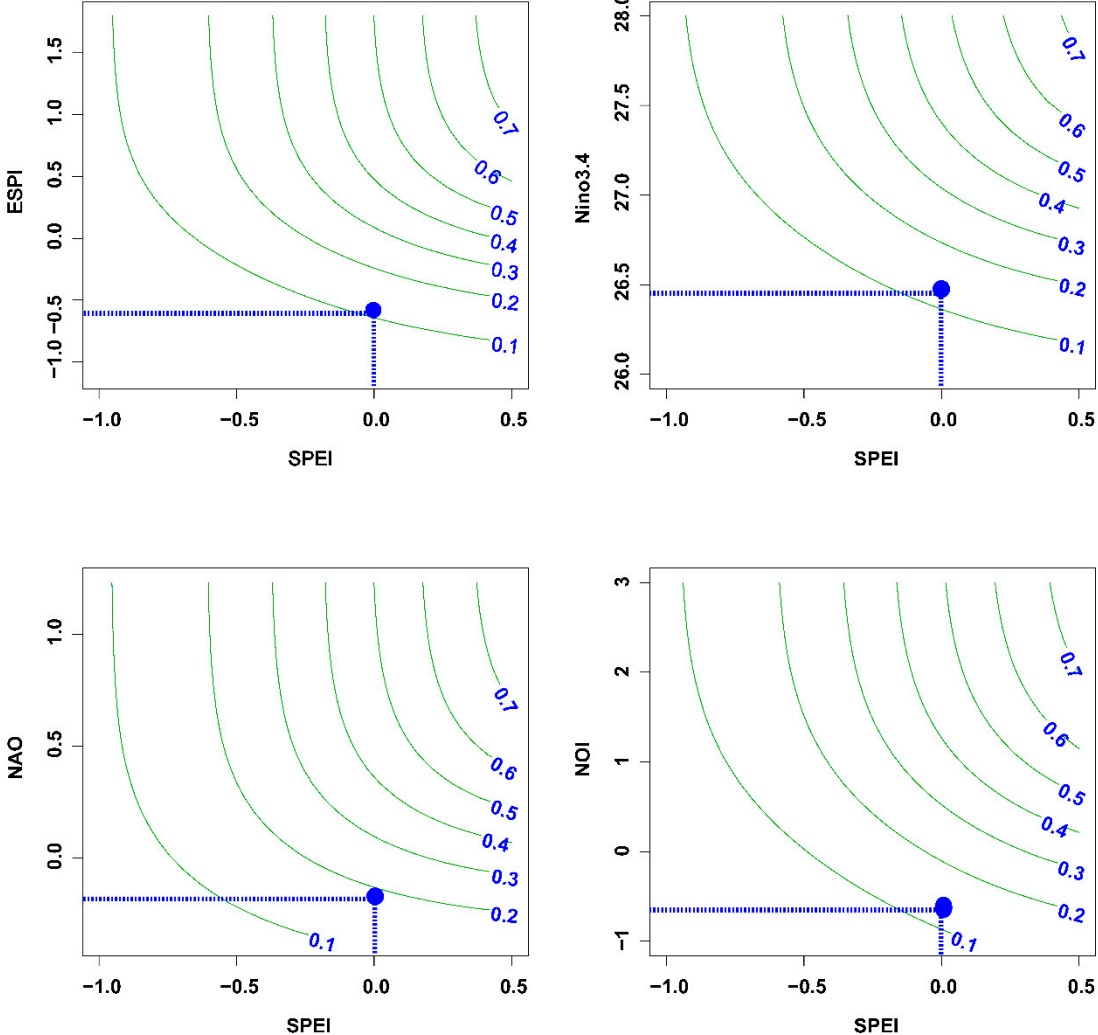

**Figure 7.** The joint probability distribution of SPEI and the GCV (when SPEI < 0). The intersection point indicates the probability of being less than the coordinate value at the same time.

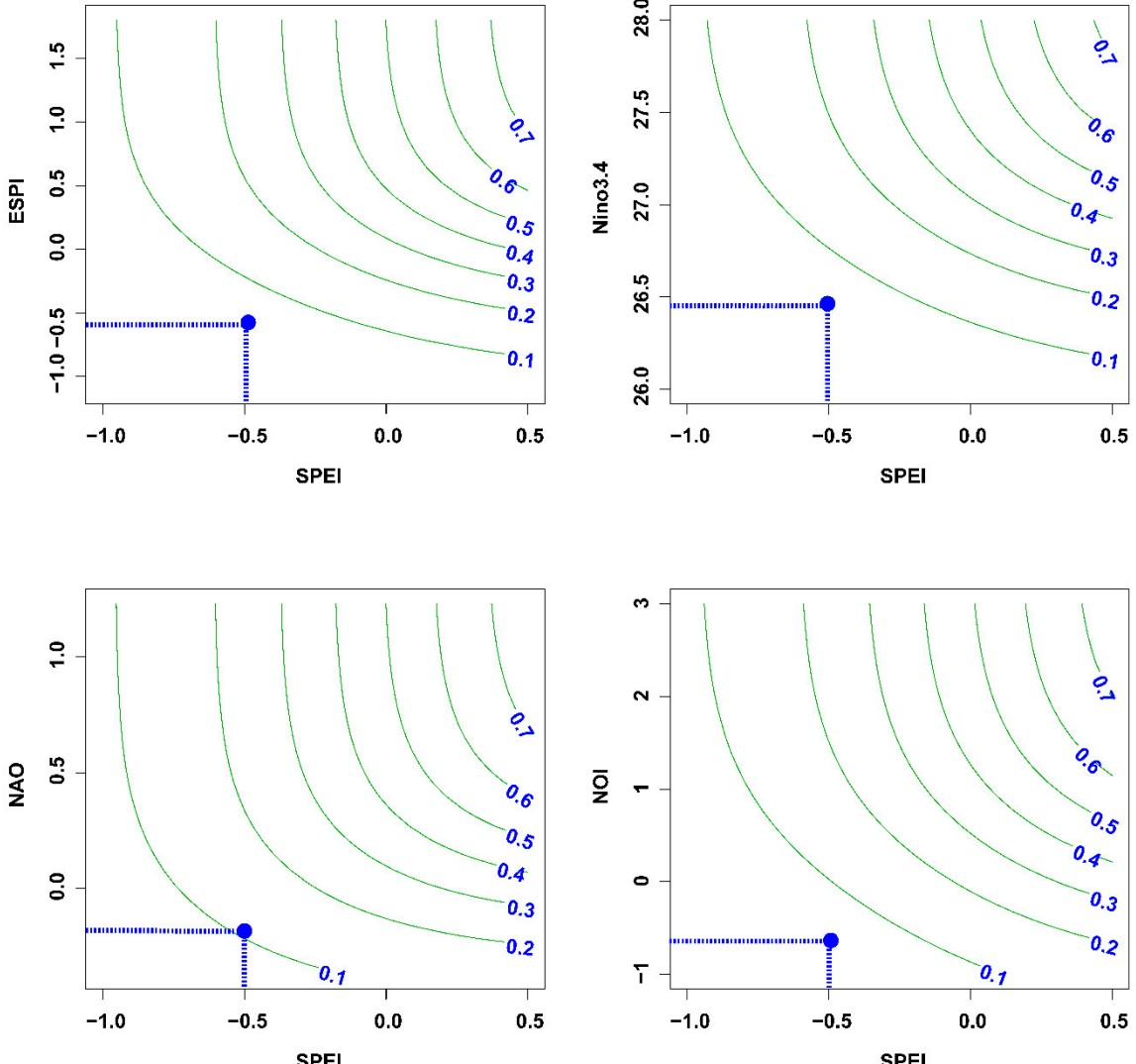

**Figure 8.** The joint probability distribution of SPEI and the GCV (when SPEI < 0.5). The intersection point indicates the probability of being less than the coordinate value at the same time.

**Table 3.** The information on the marginal distribution of variables used in this study. Location and scale are two parameters of the logistic distribution.

| Variables | Margin Distribution | Location | Scale |
|-----------|---------------------|----------|-------|
| SPEI | logistic | −0.02 | 0.43 |
| ESPI | logistic | −0.06 | 0.41 |
| NAO | logistic | 0.08 | 0.27 |
| Nino3.4 | logistic | 26.89 | 0.38 |
| NOI | logistic | 0.06 | 0.78 |

**Table 4.** The probabilities of Copula statistics.

| The GCV Indices | Threshold | SPEI | |
|---|---|---|---|
| | | **<0** | **<0.5** |
| ESPI | <1/4 | 12% | 6% |
| | >3/4 | 10% | 10% |
| Nino3.4 | <1/4 | 11% | 5% |
| | >3/4 | 12% | 12% |
| NAO | <1/4 | 16% | 9% |
| | >3/4 | 8% | 8% |
| NOI | <1/4 | 11% | 5% |
| | >3/4 | 21% | 21% |

As can be seen from Figure 8, when ESPI was less than −0.53, the concurrent probability of SPEI less than 0 was 12%; when the ESPI was greater than 0.54, the probability is 10%. For NAO, when it was less than −0.23, the probability of SPEI less than 0 was 16%, when the NAO was greater than 0.46, the probability was 8%. When the Nino3.4 was less than 26.40, the probability was 11%; when it was greater than 28.54, the probability was 12%. Lastly, when NOI was less than −0.74, the probability was 11%; when the NOI was greater than 0.85, the probability was 21%. For significant drought conditions (SPEI < −0.5), the joint probability values were relatively lower. When the four indices were smaller than their 1/4 quartiles, the probability was 6%, 9%, 5%, and 5% corresponding to ESPI, NAO, Nino3.4 and NOI respectively. When these four GCV indices were larger than their 3/4 quartiles, the joint probability was 10%, 8%, 12%, 21%, respectively. Comparing the results, it can be found that the joint probability of the NAO index was the largest when less than 1/4 quartiles. In contrast, the value of the NAO was minimal when larger than the 3/4 quartiles, and NOI was the largest. For significant drought conditions (SPEI < −0.5), the conclusions were just the same, indicating that both drought events and significant drought events have a similar connection with NAO and NOI. It can be inferred from the results that NAO was the most closely related index among these four indices, which were closely related to the drought in the YRB. It was also possible to speculate that the drought events in the YRB are more sensitive to relatively low NAO. On the other hand, when the NOI index was high, the possibility of drought events in the YRB was increasing. This was confirmed by previous studies on the influence of NAO on the East Asian monsoon in winter [67]. In the year of abnormally low NAO, the East Asian monsoon was enhanced and the precipitation in the YRB was less than that of the normal year, thus the probability of the occurrence of drought events was increased. On the contrary, the probability was reduced.

In the study of the correlation between global climate oscillations and regional climate characteristics, the dry and wet transformation mechanisms in such arid and semi-humid regions, such as the YRB, were often very complicated. At present, no physical model can capture the teleconnection between regional climate and global climate oscillation well. Therefore, the statistical model is a better means to study the climate characteristics of this complex area. As a multivariate joint probability distribution model, Copula can better reflect the potential statistical relationship among multiple variables, which has been widely applied in the field of hydrology. It also provides a new idea for exploring the inherent characteristics of global and regional climates.

The uncertainty of the statistical method comes from the possibility that different methods of Copula selection may affect the results. There are many methods for Copula selection, such as AIC, BCS, the least root mean square error, Bayesian information criterion, etc. The different selection of Copulas would result in different joint probabilities [29]. From a practical point of view, the YRB has great regional heterogeneity, and the transformation speed of drought and waterlogging is very fast. The SPEI value of the whole basin will lose a lot of information after calculating the annual average. Therefore, the seasonal characteristics of drought in the YRB are ignored. At the same time, the characteristics of drought-flood abrupt alternation in the south-east of the YRB are not well reflected. Furthermore, some updated Copula constructs such as vine Copula have been applied in research on

drought. It has an exceptionally good result [68,69]. These studies give us an indicator to use in the new methods for our further research. This study can help to identify and predict the meteorological drought and provide a feasible way for studying the response of the climate in the humid semi-humid region to the global scale climate oscillation.

## 5. Conclusions

As a typical humid and semi-humid region, the characteristics of the drought in the YRB are mainly influenced by the East Asian Monsoon and the North-West Pacific Monsoon. Together with the GCV, the spatiotemporal characteristics of the drought events in the YRB will vary accordingly. In this study, SPEI is used as an index reflecting the dry and wet condition of the YRB. The spatiotemporal distribution characteristics of SPEI values were analyzed, and a statistical model was established with nine GCV indices. Hence, the characteristics of the drought in the YRB can be identified and the characteristics of variation and quantitative assessment can be carried out. Here are the main conclusions.

The results demonstrated that there were six significant spatiotemporal modes revealed by SPEI3 (i.e., seasonal drought), which were consistent with the distribution of the main sub basins in the YRB, indicating a heterogeneity of drought regime. However, the SPEI12 (i.e., annual drought) can only reveal five modes. In the humid and semi-humid climate region, SPEI3 can better reflect seasonal scale variation. The great difference between the east and the west and the difference between the north and the south was mainly reflected as well. The periodic characteristics and correlations between SPEI and the GCV were different in the time and frequency domain. There was a strong periodic correlation between SPEI and the Precipitation Indices or the ENSO Indices in the time-frequency domain. Among GCV indices, NAO and SPEI had the same period of 8-10a and have a positive correlation. In terms of linear correlation, ESPI, NAO, Nino3.4 and NOI showed significant correlation after one year of SPEI lag, indicating that there might be a certain causal relationship. The joint probability distribution model was set up for studying the quantitative relationship between SPEI and four GCV indices, and the possibility of drought events was studied from a statistical perspective under the condition of extreme climate change. The results showed that when NAO was lower than the 1/4 quantile, the possibility of concurrent drought events reached the highest. The probability of SPEI less than 0 or less than −0.5 was 16% or 9% respectively, whereas when NAO was higher than the number of 3/4 quantile, the possibility reached the lowest. However, the result of NOI was the opposite of NAO, and the results of the other two indices were between NAO and NOI. The results above showed that the drought events in the YRB were most sensitive to the low NAO and high NOI events. NAO and NOI can be used as early warning factors for monitoring drought events. This study showed a great significance for the understanding of spatiotemporal characteristics of meteorological drought and will provide a reference for the further formulation of water resources policies and the prevention of drought disasters.

**Author Contributions:** Conceptualization, T.H. and L.X.; methodology, T.H.; software, H.F.; validation, H.F.; formal analysis, T.H.; data curation, H.F.; writing—original draft preparation, T.H.; writing—review and editing, L.X.; visualization, H.F.

**Acknowledgments:** The research was funded by the National Key R&D Program of China grant number 2018YFC0407606, Science & Technology Project of Qinghai Province grant number 2019-HZ-818, the Key Research and Development Plan of Jiangxi Province, China grant number 20171BBH80015 and the Collaborative Innovation Center of Technology and Material of Water Treatment. In addition, time series provided by the Earth System Research Laboratory (ESRL) of NOAA plays an important role in this research.

**Conflicts of Interest:** The authors declare no conflict of interest.

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
