# Peer review of "Drought Characteristics and Its Response to the Global Climate Variability in the Yangtze River Basin, China"

_water, doi:10.3390/w11010013_

Round 1

Reviewer 1 Report

This study examined the spatiotemporal characteristics of droughts in the Yangtze River Basin, China, and the relationship between drought index and several global climate indices. The methods and the results were clearly and scientifically presented. I believe that the manuscript is well written.

Please refer to the comments.

Author Response

Response to Reviewer 1 Comments

General Comments of Reviewer 1: This study examined the spatiotemporal characteristics of droughts in the Yangtze River Basin, China, and the relationship between drought index and several global climate indices. The methods and the results were clearly and scientifically presented. I believe that the manuscript is well written.

General Response: We want to express our sincere thanks to you for giving your appreciation to our manuscript and also thank you for your constructive suggestions. After carefully reviewing your comments, we have made major efforts in revising the shortcomings in the manuscript. All the revisions can be referred to in the following specific responses. We hope this manuscript has been improved after this round of review. Thank you again.

Point 1: Line 75: Put a period at the end.

Response 1: Thank you for the advice. Sorry for such a typo. A period has been added. Please refer to Line 83.

Point 2: Line 89: Legend has low resolution in Figure 1. I don’t see well.

Response 2: Thank you for the comment. Figure 1 has been replaced by a high-resolution one.

Point 3: Line 143: Fx(x) and Fx(x). Correct the sign of function.

Response 3: Thank you for the advice. Sorry for such a typo. The function has been corrected. Please refer to Line 171.

Point 4: Line 147: where c is the density function of C... Please check ‘c’ is right.

Response 4: Thank you for the comment. The function ‘c’ has been replaced. Please refer to Line 175.

Point 5: Line 331: Capitalize the first letter of ‘Clayton’.

Response 5: Thank you for the advice. The first letter of ‘Clayton’ has been capitalized. Please refer to Table 2.

Point 6: Line 365: Capitalize the first letter of ‘nino3.4’.

Response 6: Thank you for the advice. The first letter of ‘nino3.4’ has been capitalized. Please refer to Table 2.

Point 7: Line 362-382: The joint probability values were discussed. Besides this, it would better to show the results as a table if possible.

Response 7: Thank you for the advice. Yes, it is better to show the results as a table. A table has been added containing the details of the joint probability values. Please refer to Table 4.

Point 8: Line 422: Authors described as ‘The probability of SPEI less than 0 or less than –0.5 was 8% or 2%, respectively.’ However, the result for less than 0.5 was not discussed in section 4.3. Additional description is required in section 4.3.

Response 8: Thank you for the comment. For the result that the SPEI was less than -0.5, I added a description in section 4.3. ‘For significant drought conditions (SPEI<-0.5), the conclusions were just the same, indicating that both drought event and significant drought events have the similar connection with NAO and NOI.’ Please refer to Line 408. In addition, I merged Figure 7 and 9, Figure 8 and 10 for a better understanding of the result and discussion. Combing with Table 4 as you suggested, the result can be demonstrated clearly.

Reviewer 2 Report

The aim of article was drought characteristics and its response to the global climate variability in the Yangtze river basin, China. The paper in very interesting. I have read clearly and with interest the presented manuscript which demonstrates an interesting and important scientific problem in the study. The methods were chosen correctly as was the design of the research. The methods were adequately described with a sufficient review of cited literature. The results are clearly presented. I have a few comments, which I believe, can improve the work:

1. The novelty of conducted study must be emphasized in Introduction. Actually it is not visible in this part of article.

2 The table (1) should be placed in chapter 2. Actually it can cause confusion for potential reader of article.  

3.  In Study Area and Data chapter I would like to see more information about Yangtze River Basin. Please give information about basically physiographic parameters of analyzed basin (e.g. basin area, slopes, land use).

4. What about the verification regarding to obtained meteorological data. Did Authors perform a statistical verification of obtained data? What about the independence and homogeneity for analyzed time series. Did Authors study a trend in data and outliers?  

5. In lines 134-135 the Authors mentioned that Pearson correlation was used to find out the largest correlation coefficient by lagging one  year. Pearson correlation can be use, when the random variable are characterized by normal distribution. Did Authors study the distribution of this variable? If yes, what kind of test was used?

6. The name of chapter 4.3 should be start by big letter.

7. The Authors showed results obtained from Copula methods. What about uncertainty of obtained results? Did Authors considered carrying out such an analysis?   

8. The names of figures should be full. Please don’t use “same as…”

Author Response

General Comments of Reviewer 2: The aim of the article was drought characteristics and its response to the global climate variability in the Yangtze river basin, China. The paper is very interesting. I have read clearly and with interest the presented manuscript which demonstrates an interesting and important scientific problem in the study. The methods were chosen correctly as was the design of the research. The methods were adequately described with a sufficient review of cited literature. The results are clearly presented. I have a few comments, which I believe, can improve the work

General Response: We want to express our sincere thanks to you for giving your appreciation to our manuscript, especially to the part of method, design and results. Also, we want to thank you for your constructive suggestions. After carefully reviewing your comments, we have made major efforts in revising the shortcomings in the manuscript. All the revisions can be referred to in the following specific responses. We hope this manuscript has been improved after this round of review. Thank you again.

Point 1: The novelty of the conducted study must be emphasized in the Introduction. Actually, it is not visible in this part of the article. 

Response 1: Thank you for the comment. Yes, the novelty of the study is essential. Sorry for the lack of consideration for this part. Therefore, we added some descriptions in section 1. ‘Normally, the copula method was applied in the research of different drought characteristics like duration and severity [31, 34]. Considering the independence of the marginal distribution, it can be used to explore the probability of the co-occurrence of two variables that may have potential connections.’ which can be referred to in Line 68-71. ‘Studies on drought in the YRB were mainly focused on the hydrological drought, and there was not much research on the meteorological drought, which was most susceptible to the GCV. From the perspective of quantitative statistics, the research on the correlation of drought in the YRB mostly focused on the qualitative description. It is necessary to predict the probability of drought in a quantitative way.’ which can be found in Line 84-89.

Point 2: The table (1) should be placed in chapter 2. Actually, it can cause confusion for the potential reader of the article.

Response 2: Thank you for the advice. Table 1 has been placed in chapter 2. Please refer to line 118.

Point 3: In Study Area and Data chapter I would like to see more information about Yangtze River Basin. Please give information about basically physiographic parameters of the analysed basin (e.g. basin area, slopes, and land use).

Response 3: Thank you for the comment. We added more information about the Yangtze River Basin, such as the area, slopes and the land use. Also, the references have been added. Please refer to line 96-101 for more details.

Point 4: What about the verification regarding obtained meteorological data. Did Authors perform a statistical verification of obtained data? What about the independence and homogeneity for analysed time series. Did Authors study a trend in data and outliers?

Response 4: Thank you for the comment and questions. In terms of the data verification, the meteorological data in this study were verified by the previous study as mentioned in section 2. Also, a reference has been added to this description. Please refer to Line 110-111. For the question ‘What about the independence and homogeneity for analysed time series. Did Authors study a trend in data and outliers’, actually the characteristics of the time series has been analysed in the related studies as mentioned the section 1 (Line 74-75). So we believe that the data used in our study were reliable.

Point 5: In lines 134-135 the Authors mentioned that the Pearson correlation was used to find out the largest correlation coefficient by lagging one year. Pearson correlation can be used, when the random variable is characterized by a normal distribution. Did Authors study the distribution of this variable? If yes, what kind of test was used?

Response 5: Thank you for the comment and questions. Yes, the Pearson correlation can be used when the random variable is fitting a normal distribution. Actually, time series of the SPEI and the GCV indices were examined to check the assumption of normal distribution using the Kolmogorov–Smirnov test. None of the tested time series is normally distributed. Accordingly, Spearman’s and Kendall’s rank correlation coefficients were calculated to assess the relationship between the drought and the GCV. Sorry for such a typo in our manuscript. We forgot to change the description of our method in an updated version before submission. The related description has been added in section 3.3 (Line 161-164).

Point 6: The name of chapter 4.3 should start with a big letter.

Response 6: Thank you for the advice. The typo has been revised. Please refer to Line 356.

Point 7: The Authors showed results obtained from Copula methods. What about the uncertainty of obtained results? Did Authors consider carrying out such an analysis?

Response 7: Thank you for the comments. Yes, the uncertainty analysis is important for the statistical method. Actually, we put a new paragraph of the discussion on the uncertainty analysis to meet the requirement The uncertainty of the statistical method comes from the possibility that different methods of copula selection may affect the results (Line 427-439). This discussion has also been carried out on a published paper (see the reference). We hope this adjustment can answer your questions as much as possible. In future studies, we will further study more methods for uncertainty analysis.

Point 8: The names of the figures should be full. Please don’t use “same as…”

Response 8: Thank you for the advice. All the descriptions of this kind have been revised according to your suggestions.

Reviewer 3 Report

Ms: water-408431-peer-review-v1-1

Point Ms Remarks

No      Page

1.        1      The title is descriptive.

2.        1      In Abstract L22-25 a more comprehensive analysis of spatiotemporal characteristics of meteorological drought is needed.

3.        1      In Introduction L31 needs a reference (e.g. Panagoulia and Dimou, 1998). Also in L54-59 the Standardized Precipitation Evapotranspiration 49 Index (SPEI) index is selected as the method of choice. The authors admit that there is difficulty with arid climates where precipitation is near zero citing (Vicente-Serrano, S.M. et al, 2012), which however should be mentioned in Methods just as the problems with the SPEI PET equation are mentioned there.

4.        3      In Methods L103 the SPEI PET equation correction as per the Penman-Monteith method is mentioned without any scientific justification whatsoever based just on the recommendation in the FAO56 Paper which clearly is not enough. Further discussion should have been included, in particular regarding the claim of insensitivity between the Thornthwaite and Penman‐Monteith parameterizations (Schrier, Jones and Briffa, 2011).          

5.        4      In L 149 the copulas Ali-Mikhail-Haq, Clayton, Frank and Gumbel-Hougaard are reported to have been tested. No reference is made to more modern constructs such as the Erhardt–Czado construct (Erhardt and Czado, 2018) which had exceptionally good results (Bhuyan-Erhardt et al., 2019).    

6.       14     In Conclusion L 426-428, a more sound appearance of spatiotemporal characteristics of meteorological drought is needed.

Concluding Remarks: The paper requires minor corrections and clarifications.

Suggested references:

Bhuyan-Erhardt, U. et al. (2019) ‘Validation of drought indices using environmental indicators: streamflow and carbon flux data’, Agricultural and Forest Meteorology, 265(March 2018), pp. 218–226. doi: 10.1016/j.agrformet.2018.11.016.

Erhardt, T. M. and Czado, C. (2018) ‘Standardized drought indices: A novel uni- and multivariate approach’, J. R. Stat. Soc. Ser. C Appl. Stat., 67, pp. 643–664. Available at: http://arxiv.org/abs/1508.06476.

Panagoulia D., and Dimou G., DEFINITIONS AND EFFECTS OF DROUGHTS In Conference on Mediterranean Water Policy: building on existing experience, Mediterranean Water Network. Valencia, Spain, Volume: I, General lecture, invited presentation, ResearchGate, April 1998.

Schrier, G. Van Der, Jones, P. D. and Briffa, K. R. (2011) ‘The sensitivity of the PDSI to the Thornthwaite and Penman ‐ Monteith parameterizations for potential evapotranspiration’, JOURNAL OF GEOPHYSICAL RESEARCH, 116, pp. 1–16. doi: 10.1029/2010JD015001.

Vicente-Serrano, S.M.; Begueria, S.; Lorenzo-Lacruz, J.; Camarero, J.J.; Lopez-Moreno, J.I.; Azorin-

Molina, C.; Revuelto, J.; Moran-Tejeda, E.; Sanchez-Lorenzo, A. Performance of drought indices for

ecological, agricultural, and hydrological applications. Earth Interact 2012, 16.

Author Response

General Response: We want to express our sincere thanks to you for your constructive suggestions. After carefully reviewing your comments, we have made major efforts in revising the shortcomings in the manuscript. All the revisions can be referred to in the following specific responses. We hope this manuscript has been improved after this round of review. Thank you again.

Point 1: The title is descriptive. 

Response 1: Thank you for the comment.

Point 2: In Abstract L22-25 a more comprehensive analysis of spatiotemporal characteristics of meteorological drought is needed.

Response 2: Thank you for the advice. According to your suggestions, we added a more comprehensive description in this part. ‘The results demonstrated that there were 6 significant spatiotemporal modes revealed by SPEI3 (i.e. seasonal drought), which were consistent with the distribution of the main sub-basins in the YRB, indicating a heterogeneity of drought regime. However, the SPEI12 (i.e. annual drought) can only reveal 5 modes.’ which can be found in Line 18-21.

Point 3: In Introduction, L31 needs a reference (e.g. Panagoulia and Dimou, 1998). Also in L54-59, the Standardized Precipitation Evapotranspiration 49 Index (SPEI) index is selected as the method of choice. The authors admit that there is difficulty with arid climates where precipitation is near zero citing (Vicente-Serrano, S.M. et al, 2012), which however should be mentioned in Methods just as the problems with the SPEI PET equation are mentioned there.

Response 3: Thank you for the comment and advice. An essential reference has been added according to your advice in Line 35. Also, we added some descriptions in section 3.1 for the SPEI. ‘ The previous study admitted that there was a difficulty with arid climates where precipitation is near zero, considering that the SPEI is sensitive to precipitation [28]. Due to the humid climate in the YRB, the SPEI is suitable in this study’ (Line 122-125).

Point 4: In Methods L103 the SPEI PET equation correction as per the Penman-Monteith method is mentioned without any scientific justification whatsoever based just on the recommendation in the FAO56 Paper which clearly is not enough. Further discussion should have been included, in particular regarding the claim of insensitivity between the Thornthwaite and PenmanMonteith parameterizations (Schrier, Jones and Briffa, 2011).

Response 4: Thank you for the comment. We added a description of the strength and shortcoming of this two method. ‘The empirical Thornthwaite equation [43] and the Penman-Monteith equation are the most popular methods [44,45] used to calculate PET. The empirical Thornthwaite equation is considered to be most convenient because the air temperature is the only variable in the formula. For the assessment of meteorology drought, this method provides a fast way to obtain a PET. When there are more observation data such as wind speed, cloud amount and relative humidity etc., the Penman-Monteith equation is a better choice due to its consideration of energy balance and atmospheric water demand [46].’ Which can be found in Line 125-131. We hope this part of the discussion can meet the requirement.

Point 5: In L 149 the copulas Ali-Mikhail-Haq, Clayton, Frank and Gumbel-Hougaard are reported to have been tested. No reference is made to more modern constructs such as the Erhardt–Czado constructs (Erhardt and Czado, 2018) which had exceptionally good results (Bhuyan-Erhardt et al., 2019).

Response 5: Thank you for the comment. Yes, the method in the reference you mentioned above is a state-of-the-art method, which is well worth learning. We conducted this study referring to the previous studies using the traditional copula functions and we want to do further research by applying modern constructs. Actually, we added some content in section 4.3 (Line 435-436). And the related references have been added.

Point 6: In Conclusion L 426-428, a more sound appearance of spatiotemporal characteristics of meteorological drought is needed.

Response 6: Thank you for the advice. We added the related descriptions in this part (Line 449-454).

Reviewer 4 Report

The paper analyzes the frequency of occurrence of droughts in the Yangtze River Basin in relation to the global climate variability (GCV), characterized by selected indices. In my opinion, the submission is interesting and deserves attention, especially if consider the projected climate change scenarios for that area. However, there exist some shortcomings, which require improvements prior to the final acceptance of the submission for publication. They are as follows:

1. In the description of the indices applied in the study there is a lack of important details, for example, regarding NAO – is it the annual NAO or the winter NAO? Also, the method of its calculation is missing.

2. Figure 6 – the caption is unclear, so it is suggested to change it. Also, the meaning of the red line is not clear: “Red dotted lines represent significance” – but does it mean a CHANGE in the significance after the 1-year lag was applied (the original not-significant values became significant after the 1-year lag)? Please explain.

3. As I understand the Authors assumed that results given in Table 2 are the bases for selection of the best-fitted copulas. Actually, the AKAIKE criterion is commonly applied in the selection procedure. Please explain.

4. What is the meaning of the values given in Table 3? Are they the location and scale PARAMETERS?

5. Figures 7-10 – the letters “a”, “b”, “c” and “d” are not explained. In fact, they seem not to be necessary. If so, it is suggested to delete them.

6. In my opinion Figures 7 and 8 may be merged into one figure containing four graphs. The same refers to Figures 9 and 10.

7. Figures 7-10 – what is the reason of drawing the lines using two different colors (red and dark blue)?

8. Figures 7-10 – the interval marks on the horizontal axes referring to ESPI are not shown. Please correct.

9. Figure 8 – caption: “Same as Figure 7, but the joint probability is equal to the value of the marginal probability of SPEI at 0 minuses the intersection point value”. What is the meaning of “0 minuses”? It is quite confusing.

10. Page 11, l. 342: “The 25% and 75% quantiles were used to define the global extreme climate (P. Shi et al.)”. What do the Authors mean by “the global extreme climate”? Besides “Shi et al.” cannot be found in References.

11. Page 13, l. 362-363: “As can be seen from Figure 8, when ESPI was less than -0.53, the concurrent probability of SPEI less than 0 was 12%; when the ESPI was greater than 0.54, the probability is 10%”. In my opinion this statement does not fully correspond with the results seen on Figure 8. So, it would be helpful to explain shortly how the values of 12% and 10% were obtained.

12. The English language requires corrections.

Author Response

General Comments of Reviewer 4: The paper analyses the frequency of occurrence of droughts in the Yangtze River Basin in relation to the global climate variability (GCV), characterized by selected indices. In my opinion, the submission is interesting and deserves attention, especially if consider the projected climate change scenarios for that area. However, there exist some shortcomings, which require improvements prior to the final acceptance of the submission for publication. They are as follows:

General Response: We want to express our sincere thanks to you for giving your appreciation to our manuscript. Also, we want to thank you for your constructive suggestions. After carefully reviewing your comments, we have made major efforts in revising the shortcomings in the manuscript. All the revisions can be referred to in the following specific responses. We hope this manuscript has been improved after this round of review. Thank you again.

Point 1: In the description of the indices applied in the study there is a lack of important details, for example, regarding NAO – is it the annual NAO or the winter NAO? Also, the method of its calculation is missing.

Response 1: Thank you for the comment. We added the description of the details as you required in section 2. The original data of the indices were all monthly data. An average calculation was applied to obtain an annual one (Line 113-114).

Point 2: Figure 6 – the caption is unclear, so it is suggested to change it. Also, the meaning of the red line is not clear: “Red dotted lines represent significance” – but does it mean a CHANGE in the significance after the 1-year lag was applied (the original not-significant values became significant after the 1-year lag)? Please explain.

Response 2: Thank you for the comment. The caption of Figure 6 has been changed. The red line in the figure has been deleted. We use a better way to demonstrate the result of lag analysis only using a solid and a hollow triangle. Two adjacent triangles were grouped, with the first one representing the original sequence and the second one representing the lagging sequence. Please refer to Line 306.

Point 3: As I understand the Authors assumed that results given in Table 2 are the bases for selection of the best-fitted copulas. Actually, the AKAIKE criterion is commonly applied in the selection procedure. Please explain.

Response 3: Thank you for the comment. Yes, the method used in our study was just the AICc (a calibrated AIC method) method. The p-value mentioned in Table 2 was just the AICc value. Sorry for the lack of information. We added the details of AICc in section 3.4 (Line 178-183).

Point 4: What is the meaning of the values given in Table 3? Are they the location and scale PARAMETERS?

Response 4: Thank you for the question. Yes, the location and scale are the two parameters in the logistic distribution.

Point 5: Figures 7-10 – the letters “a”, “b”, “c” and “d” are not explained. In fact, they seem not to be necessary. If so, it is suggested to delete them.

Response 5: Thank you for the advice. We agree with this point. The letters have been deleted from the figures.

Point 6: In my opinion Figures 7 and 8 may be merged into one figure containing four graphs. The same refers to Figures 9 and 10.

Response 6: Thank you for the advice. After consideration, we merged the four figures mentioned above into 2 figures only containing the information when the GCV indices are less than their 1/4 quantiles. In addition, table 4 has been added following the figures containing the details of the probability when the GCV indices are larger than their 3/4 quantiles. We believe that this revision can better demonstrate the results. Please refer to the new version of figure 7 and 8.

Point 7: Figures 7-10 – what is the reason for drawing the lines using two different colours (red and dark blue)?

Response 7: Thank you for the question. Actually, we realized that it was indeed unnecessary to draw the lines using two different colours. As we revised the figures according to Point 6, we change the colour to a single one as well. Please refer to the new version of figure 7 and 8.

Point 8: Figures 7-10 – the interval marks on the horizontal axes referring to ESPI are not shown. Please, correct.

Response 8: Thank you for the comment. The error has been corrected.

Point 9: Figure 8 – caption: “Same as Figure 7, but the joint probability is equal to the value of the marginal probability of SPEI at 0 minuses the intersection point value”. What is the meaning of “0 minuses”? It is quite confusing.

Response 9: Thank you for the comment. We added a detail in section 3.4 (Line 184-187). Please refer to equation (7). Also, the details of the probability can be found in the new Table 4.

Point 10: Page 11, l. 342: “The 25% and 75% quantiles were used to define the global extreme climate (P. Shi et al.)”. What do the Authors mean by “the global extreme climate”? Besides “Shi et al.” cannot be found in References.

Response 10: Thank you for the comment. Actually, the 1/4 and 3/4 quantiles were used to define the extreme GCV. The purpose of establishing the joint probability distribution was to explore the response of flood-drought events to extreme GCV. “the global extreme climate” may cause the confusion, so we changed the description to the extreme GCV. Please refer to Line 376. The reference here is a typo, sorry for our mistake. We have deleted it.

Point 11: Page 13, l. 362-363: “As can be seen from Figure 8, when ESPI was less than -0.53, the concurrent probability of SPEI less than 0 was 12%; when the ESPI was greater than 0.54, the probability is 10%”. In my opinion, this statement does not fully correspond with the results seen in Figure 8. So, it would be helpful to explain shortly how the values of 12% and 10% were obtained.

Response 11: Thank you for the comment. Actually, the response 9 can also help to explain this confusion. We added a detail in section 3.4 (Line 184-187). Please refer to equation (7). Also, the details of the probability can be found in the new Table 4. I believe this kind of confusion can be eliminated with the details we provided.

Point 12: The English language requires corrections.

Response 12: Thank you for the suggestion.  The manuscript was checked by a native English speaking colleague. I hope the revised version can meet the requirement.
